# Insights into structure and dynamics of extracellular domain of Toll-like receptor 5 in *Cirrhinus mrigala* (mrigala): A molecular dynamics simulation approach

**Ajaya Kumar Rout**[1☯], **Varsha Acharya**[1], **Diptimayee Maharana**[1], **Budheswar Dehury**[2☯], **Sheela Rani Udgata**[3], **Rajkumar Jena**[4], **Bhaskar Behera**[4], **Pranaya Kumar Parida**[1]*, **Bijay Kumar Behera**[1]*

1 Aquatic Environmental Biotechnology & Nanotechnology (AEBN) Division, ICAR-Central Inland Fisheries Research Institute, Kolkata, West Bengal, India, 2 Department of Chemistry, Technical University of Denmark, Kongens Lyngby, Denmark, 3 Department of Bioinformatics, Odisha University of Agriculture and Technology, Bhubaneswar, Odisha, India, 4 Department of Biosciences and Biotechnology, Fakir Mohan University, Balasore, Odisha, India

☯ These authors contributed equally to this work.
* beherabk18@yahoo.co.in (BKB); pranayaparida@gmail.com (PKP)

**Data Availability Statement:** All relevant data are within the paper and its Supporting information files.

## Abstract

The toll-like receptor 5 (TLR5) is the most conserved important pattern recognition receptors (PRRs) often stimulated by bacterial flagellins and plays a major role in the first-line defense against invading pathogenic bacteria and in immune homeostasis. Experimental crystallographic studies have shown that the extracellular domain (ECD) of TLR5 recognizes flagellin of bacteria and functions as a homodimer in model organism zebrafish. However, no structural information is available on TLR5 functionality in the major carp *Cirrhinus mrigala* (mrigala) and its interaction with bacterial flagellins. Therefore, the present study was undertaken to unravel the structural basis of TLR5-flagellin recognition in mrigala using structural homodimeric TLR5-flagellin complex of zebrafish as reference. Integrative structural modeling and molecular dynamics simulations were employed to explore the structural and mechanistic details of TLR5 recognition. Results from structural snapshots of MD simulation revealed that TLR5 consistently formed close interactions with the three helices of the D1 domain in flagellin on its lateral side mediated by several conserved amino acids. Results from the intermolecular contact analysis perfectly substantiate with the findings of per residue-free energy decomposition analysis. The differential recognition mediated by flagellin to TLR5 in mrigala involves charged residues at the interface of binding as compared to the zebrafish complex. Overall our results shows TLR5 of mrigala involved in innate immunity specifically recognized a conserved site on flagellin which advocates the scientific community to explore host-specific differences in receptor activation.

**Funding:** The author(s) received no specific funding for this work.

**Competing interests:** The authors have declared that no competing interests exist.

## Introduction

The innate immune system provides a robust and efficient first line of defense against invading microbial pathogens through evolutionary conserved pattern recognition receptors (PRRs) [1]. These conserved PRRs are broadly classified into four categories *viz.* NOD-like receptors (NLRs), RIG-I-like receptors (RLRs), C-type lectin receptors (CLRs), and toll-like receptors (TLRs) [2]. Among these PRRs, TLRs are germline-encoded receptors which not only elicit defense response against invasion of pathogens, but also trigger rapid inflammatory responses upon recognition of ligands. TLRs are comprised of three domains *i.e.* the extracellular domain (ECD), transmembrane domain (TMD) and the intracellular domain. Leucine-rich repeats (LRRs) forms the N-terminal ECD, often recognize a number of pathogen-associated molecular patterns (PAMPs) following signal transduction to the cytoplasmic components *via* middle single-pass TMD, while intracellular Toll/IL-1 receptor (TIR) domain responsible for instigating the signal transduction [3]. TLRs can recognize numerous ligands *i.e.*, molecules derived from pathogens (exogenously) and processed by the host cell (endogenously).

In mammals, a total of 13 TLRs have been reported, of which 10 are reported in cattle, buffalo, sheep, pig, and human [4, 5]. Depending upon their localization inside the host cell, TLRs can be categorized into two groups [6]. TLR1, TLR2, TLR4, TLR5, TLR6, and TLR10 form the first group, mostly express in cell surface and recognize bacterial derived molecules. While the other category *i.e.* TLR3, TLR7, TLR8, and TLR9 typically express on the membranes of intracellular organelles, recognizes nucleic acids and nucleotide derivatives of viral and bacterial origins [7]. Among the two classes of TLRs, TLR5 acts as a major determinant of host-pathogen interaction and immune homeostasis [8]. In vertebrates, bacterial flagellins are the molecular stimuli that ligate and activate TLR5. The recognition process of bacteria by TLR5 has been linked with many non-infectious diseases, especially in gastrointestinal tract of vertebrates, such as shaping the gut microbiota, metabolic tolerance, and immune balance [9].

The structural protein flagellin accumulates into a filament of flagellum which expands from the surface of bacterial cell and permits bacteria to become motile [10]. These structural abundant proteins also promote the invasion and adhesion of pathogenic bacteria into host cells, often represents central target of immune surveillance in the host cell. Upon bacterial invasion, flagellin is detected by host TLRs *i.e.* TLR5 and NLRC4, and triggers first line of defense thereby pave the way for removal of pathogens [11, 12].

The flagellins in bacteria are comprised of two to four domains. *Bacillus subtilis* Hag flagellin comprised of two domains (D0 and D1), *Pseudomonas aeruginosa* type A FliC flagellin comprised of three domains (D0, D1, and D2), and while *Salmonella enterica* subspecies enterica serovar Typhimurium FliC flagellin possess four domains (D0, D1, D2, and D3). The evolutionarily conserved D0 and D1 domains are found to be concealed in the central part of flagellar filament supporting inter-flagellin interactions in bacteria [13–15]. Both domains of flagellin monomer act as the vital stimulator of TLR5 [16]. However, in case of flagellins formed of three- and four-domains, D1 is connected to D2, and D3 has little or no role in filament formation. In addition, compared to D0 and D1, the D2 or D3 domains display a considerable difference at both sequence and structural level and are believed to trigger adaptive immunity, often results in adverse toxicity to the therapeutics mediated by flagellin. In some bacteria like *Clostridium difficile* and *Bacillus subtilis*, flagellin lacks hypervariable domains and harbors only D0 and D1 domain needed for TLR5 activation and flagellin polymerization [17].

Over the last decade, several studies have shown the interaction of TLR5-flagellin and its cellular outcome using *Salmonella flagellins*. Recent structural and biochemical study of a complex between the N-terminal fragment of zebrafish TLR5 and *Salmonella enterica* subspecies

*enterica serovar* Dublin flagellin revealed that flagellin and TLR5 form a 1:1 complex through primary interaction and upon homo-dimerization forms 2:2 complex [18, 19]. It is still under debate that how numerous flagellins follow the TLR5-recognition mechanism observed for *Salmonella flagellin* despite variations in the sequences and domains of flagellins.

TLRs have also been reported in fishes which share close homology with eukaryotes including humans [20, 21]. In teleosts, more than 20 TLRs have been discovered underlining their significance in providing the first line of defense in fishes. TLR6 and TLR10-13 are absent, while, multiple copies of TLRs (*i.e.* TLR3, TLR4, TLR5, TLR7, TLR8, TLR20 and TLR22) exists which are reported to be involved in the development of fish [21]. TLR1-13, are found be mutual between humans and fish which may have different ligands and functions in fish and mammals [8, 22–24]. Upon cleavage of the signal peptides, the matured fish TLRs formed of ~800–900 amino acids, where, the ECD comprise of variable number of LRRs. These LRRs often share considerable identity with TLRs from zebrafish, human, and mouse at sequence level. Significant variations are also observed in the number of LRR repeats across species and TLR class. Moreover, the variation in composition of LRR motifs across TLRs makes them selective to recognize a target PAMP [25]. Apart from the ECD, the TMD helix comprised of ~20 amino acids which bridge the cytoplasmic TIR domain with the ECD. Among these three domains, TIR domain is relatively strongly conserved in fish and mammals as compared to the LRR domain [26].

In fish, TLRs are the most studied PRRs and their PAMP selectivity has been reported extensively [27]. The bacterial components like peptidoglycans (PGNs), lipoteichoic acid (LTA), lipopolysaccharides (LPS), and flagellin are recognized by fish TLRs. Most interestingly, in higher eukaryotes, both LPS and flagellins are recognized by TLR4 and TLR5, however, in some fish species, due to lack of TLR4, they do not recognize LPS; however TLR5 possess multiple copies depicting the selectivity towards PAMP [8, 28–30]. TLR3, TLR9, TLR13, and TLR22 have shown to interact with bacterial and viral RNA, while PMAPs for TLR1, TL7, TLR8, TLR13-20, and TLR23-27 are underexplored [31–35].

In fishes, TLR-ECD is comprised of the LRR domains which are unstructured in the convex surface and appear like a horseshoe structure. The hetero-dimeric structure of LTR5a-TLR5b reported in zebrafish [8] shares similar architecture to that of the crystallographic structure of TLR-Flagellin complex [18]. Due to the lack of biochemical and structural information of TLRs in fishes, the knowledge of TLR mediated flagellin recognition is poorly understood. The lack of this crucial structural insight hinders development of new flagellin-based therapeutics. Therefore, in the present study, we made an attempt to investigate the mechanism by which flagellins recognize TLR5 in the major carp mrigala (*Cirrhinus mrigala*) using experimental TLR5-flagellin complex [18] from zebrafish as a template. Integrative modeling and extensive long-term all-atoms molecular dynamics simulations were conducted to infer the plausible mode of flagellin-mediated TLR5 interaction and the binding site interface. Furthermore, our study highlights the energetic contribution which drives the binding of flagellin to TLR5 and the importance of residues that aid in molecular recognition.

## Materials and methods

### *In silico* analysis of TLR5 from *Cirrhinus mrigala*

The primary amino acid sequence of *Cirrhinus Mrigala* TLR5 (*Cm*TLR5) was obtained from the UniProt database (ID: W6AYY9). Protein family, domains and motif information were inferred using Pfam [36], SMART [37], and CD-search [38] tools. The PSIPRED server [39] was used to predict the secondary structure from the primary amino acid sequence.

## Computational modeling of the ECD of *Cm*TLR5

Due to a lack of structural information on *Cm*TLR5, we employed theoretical modeling approach to predict the three-dimensional (3D) structure of TLR5-ECD (Lys20-Ser424 region). Initially, templates were obtained by performing BLASTp [40] against the protein data bank (PDB). The BlastP result revealed the crystal structure (PDB ID: 3V47_A) of the N-terminal fragment of zebrafish TLR5 as the most reliable template for protein modeling. Modeller version 9.21 [41] was employed to build discrete 3D structures based on the sequence alignment of *Cm*TLR5 and template 3V47 (A-chain). 3D protein models were sorted according to their DOPE score, where models with the least DOPE score was selected for additional optimization. The side chains were optimized using WHAT-IF [42]. The final 3D protein model was evaluated to test the stereo-chemical quality of the model through various protein model evaluation tools including PROCHECK [43], ERRAT [44], ProSA [45], and ProQ [46].

## Generation of *Cm*TLR5-Salmonella flagellin complex

To understand the mode of flagellin mediated TLR5 interaction, the coordinates of flagellin from structural homolog 3V47 was transferred to modelled ECD of TLR5 upon structural superposition in PyMOL (The PyMOL Molecular Graphics System, Version 2.0 Schrödinger, LLC.). The protocol for transfer of coordinates was implemented from earlier studies reported elsewhere [47–49]. Resultant complex was checked to remove bad contacts or bumps using BIOVIA Discovery Studio version 4.5 (BIOVIA DSV). To understand the structural dynamics of modeled *Cm*TLR5 and *Cm*TLR5-flagellin complex, long-term molecular dynamics (MD) simulations were performed in GROMACS. For cross-comparison, we also performed an MD simulation of the experimental *Dr*TLR5-flagellin complex (3V47) using the same protocol as described below.

## Molecular dynamics simulations

All-atoms MD simulations of the three systems *i.e.*, *Cm*TLR5 (alone), *Cm*TLR5-flagellin, and *Dr*TLR-flagellin complexes were conducted using Amber ff99SB-ILDN force field [50] in GROMACS version 2018.4 [51]. Each system was solvated using TIP3P water models in a cubic simulation box with a salt (NaCl) concentration of 0.15 M, where the box size is defined by a minimum of 10 Å from the box edge to each protein/complex. To remove bad contacts or clashes, each electro-neutralized system was subjected to steepest-descent energy minimization followed by two step equilibration through NVT and NPT ensemble. All the systems were equilibrated at 300 K and 1 atm using 1 ns in the NVT ensemble, and 1 ns in the NPT ensemble respectively. Long-range electrostatic interactions were treated with the Particle Mesh Ewald algorithm, while, all bonds were constrained using the Linear Constraint Solver (LINCS) algorithm. Finally, after two-step equilibration, each system was subjected to production MD for 200 ns. The parameters and procedures of MD simulation were adopted from earlier studies [52–56]. The trajectory files of each system were analyzed using VMD [57] and built-in modules of GROMACS. The 2D plots depicting the dynamics stability of each system including the backbone root mean squared deviation (RMSD), radius of gyration ($R_g$), solvent accessible surface area (SASA), and inter-molecular hydrogen bonds were plotted through XmGrace 5.1.23 software. The inter-molecular contacts were analyzed using BIOVIA DSV and PyMOL.

## Essential dynamics

To understand the collective motions in each system, we performed principal component analysis (PCA) on the main-chain atoms using *gmx anaeig* and *gmx covara* tool of GROMACS

by considering the last 100 ns trajectory from each system [58]. PCA approach was employed to calculate eigenvectors (EVs) or principal components (PCs), and eigenvalues and their projection along with the first two PCs. As the first few PCs explain the total variance, we only explored the top two PCs *i.e.*, PC1 and PC2 to explain the global motions of TLR5 (alone), *Cm*TLR5, and *Dr*TLR5 complexes.

## Calculation of binding free energy

The molecular mechanics/Poisson–Boltzmann surface area (MM/PBSA) approach is a method of choice to estimate free energy of binding of macromolecular complexes including protein-protein, protein-peptides and receptor-ligands [59]. Therefore, to estimate the binding free energy of the *Cm*TLR5-flagellin and *Dr*TLR5-flagellin complexes from MD trajectories, we employed the MM/PBSA approach implemented in g_mmpbsa tool. For computing the free energy of binding, we used 300 snapshots from the last 100 ns trajectories of both the systems at equal intervals of time [52, 60, 61].

## Results and discussion

### Molecular evolutionary analysis of *Cm*TLR5

To understand the molecular evolution of TLR5 from *Cirrhinus mrigala*, we performed a phylogenetic analysis of *Cm*TLR5 with the sequences of TLR5 from closely related organisms obtained after BLASTp search against the non-redundant database of NCBI. The 2-dimensional phylogenetic tree was constructed using the Neighbor-Joining method with an iteration of 1000 times using MEGA6 [62]. The optimal tree with the sum of branch length of 9.07 has been displayed in Fig 1. The evolutionary analysis was involving 73 sequences. Altogether, four major clusters were observed, where TLR5 from *Cirrhinus mrigala*, *Carassius auratus*, *Schizothorax richardsonii*, *Sinocyclochellus rhinocerous*, and *Sinocyclocheilus grahami* formed a major cluster (cluster 3). However, *Danio rerio* along with *Ctenopharyngodon idella*, *Sinocyclocheilus anshuiensis*, and *Cyprinus carpio* formed another clade in the same cluster 3. From the phylogenetic tree, it can be observed that TLR5 from *Cirrhinus mrigala* forms major clades with other fish species and model organism zebrafish.

### Major domains in *Cm*TLR5

TLRs in fishes share a higher percentage of sequence homology with other eukaryotes including humans. As compared to humans, fishes possess more TLRs as they are exposed to an array of microbes. In contrast, 13 TLRs have been reported in mammals, while, in case of teleosts, more than 20 TLRs were identified signifies the implication of these TLRs in innate immunity in fish [27]. Among these TLRs in fishes, TLR5 possess multiple copies and are associated with the development process. Across species, the number of LRR repeats varies and the composition of amino acids in these sequence motifs varies across the TLRs which makes them selective towards specific PAMPs [25]. Primary sequence analysis of *Cm*TLR5 through CD search and SMART revealed three-domain architecture *i.e.* the ECD (20–640 amino acids) comprised of 16 conserved LRR repeats, the middle transmembrane domain (653–675), and the TIR domain (705–852). To understand the variations and similarity in the sequences of TLR5-ECD in diverse species, we also performed a multiple sequence alignment of *Cm*TLR5-ECD with its close homologs (S1 Fig in S1 File). We observed profound variation over the entire stretch of ECD which mostly covers the LRR repeats which perfectly correlates with the earlier studies of TLRs in fishes [27].

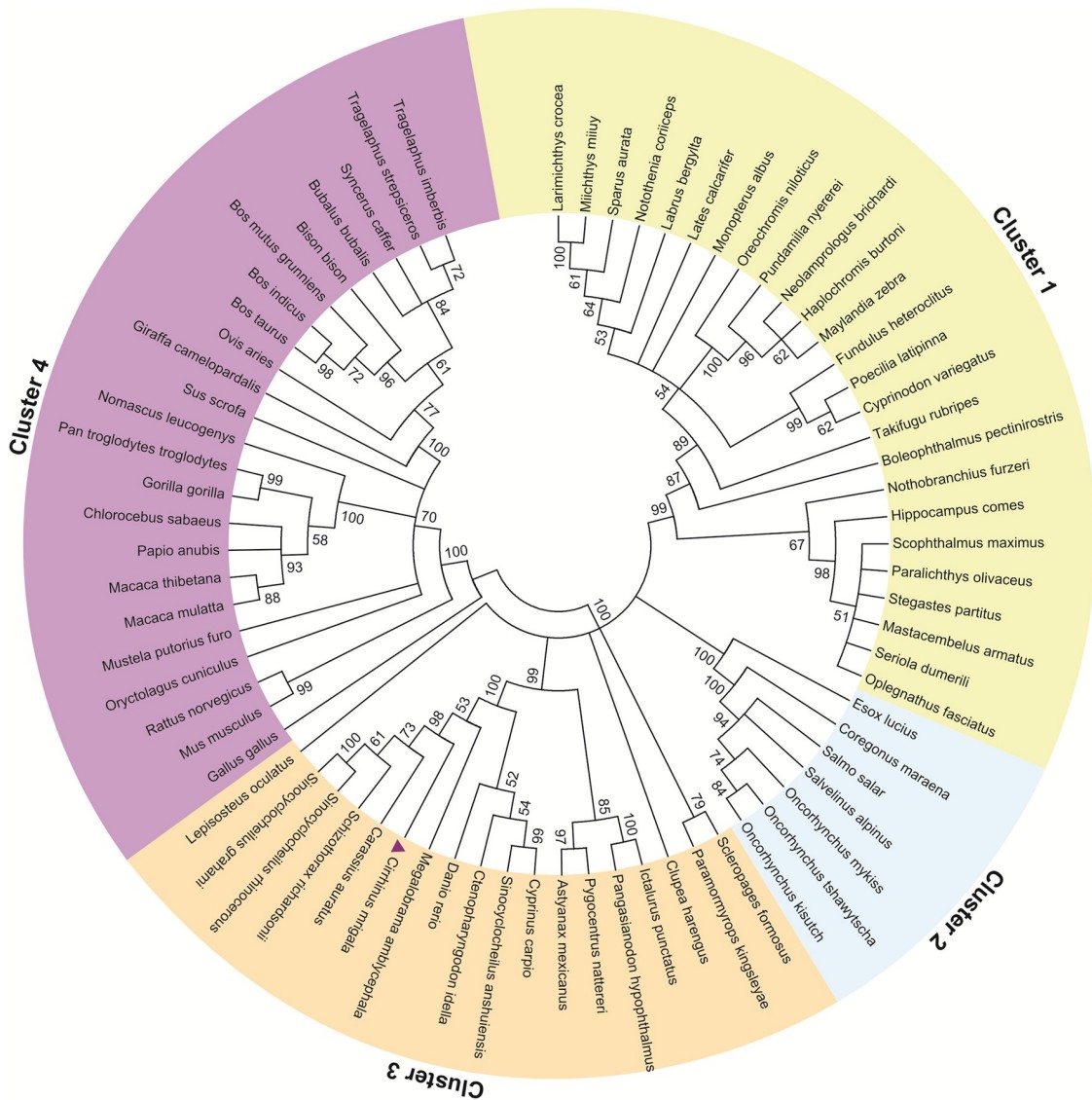

**Fig 1. Evolutionary relationship of taxa using TLR5 from *Cirrhinus mrigala* and other closely related organisms using NJ method.** The percentage of replicate trees in which the associated taxa clustered together in the bootstrap test (1000 replicates) was displayed. Evolutionary analyses were conducted on MEGA 6.0.

## Modeling of ECD *Cm*TLR5

Like other TLRs in fishes, TLR5 of *Cirrhinus mrigala* comprised of 877 amino acids long and often recognize bacterial component flagellin. To understand the mode of flagellin recognition in *Cm*TLR5 ECD, we employed a combinatorial approach involving computational modeling, molecular dynamics simulation, and binding free energy calculations. Due to a lack of structural information on *Cm*TLR5, we modelled the LRR region encompassing Lys20-Ser424 (the other parts are not modeled due to lack of structural information from the template). Suitable templates for computational modeling of the LRR region of *Cm*TLR5 were identified through BLASTp and DELTA-BLAST search. In consensus, both suggested the N-terminal fragment of zebrafish TLR5 (PDB ID: 3V47) as the most reliable template with a sequence similarity of 74%. Modeller aided in the development of the raw protein models, considering the pair-wise

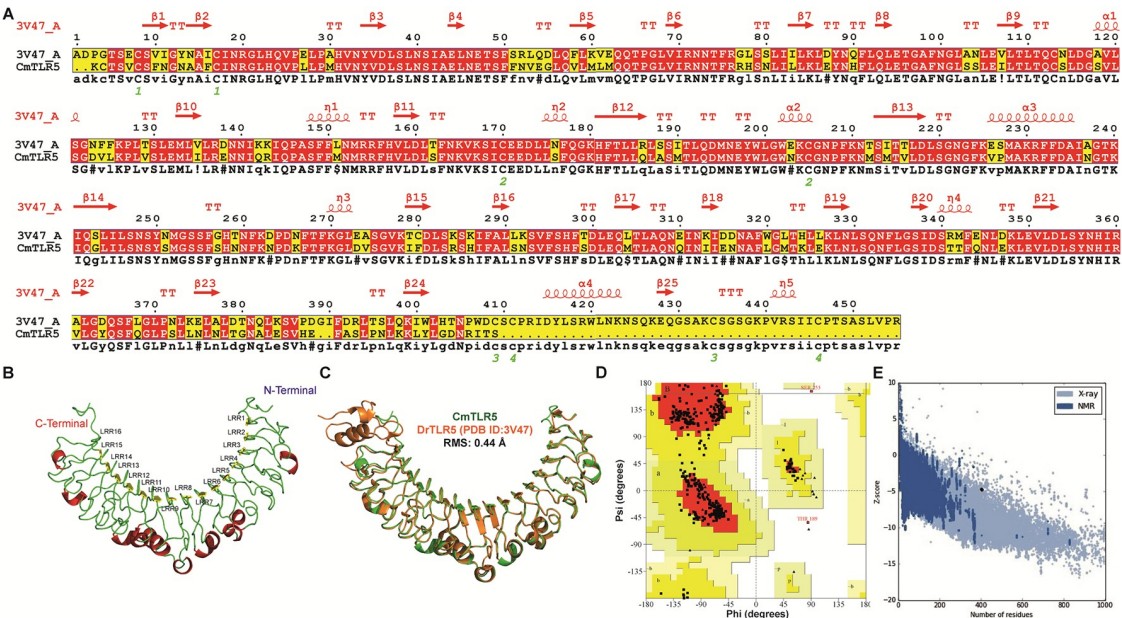

**Fig 2. Sequence-structure alignment, modeling, and validation of the modelled *Cm*TLR5.** (**A**) Target-template alignment of *Cm*TLR5 with the close structural homolog 3V47 using Multalin and ESPript. (**B**) The architecture of the modelled *Cm*TLR5 protein displaying the sixteen LRRs and variable loops. (**C**) Structural superimposed view of the modeled *Cm*TLR5 concerning the template *i.e.* TLR5 from zebrafish. (**D**) Ramachandran plot of the modeled *Cm*TLR5 was obtained using PROCHECK. (**E**) Evaluation of the modeled *Cm*TLR5 through z-score using ProSA-web.

sequence alignment (Fig 2A) of *Cm*TLR5 and the crystal structure of the homolog. Among the models, the protein model with the least DOPE score and least RMSD as compared to the template was selected for model optimization.

## Topology of the modelled *Cm*TLR5

Like the structural homolog, the modeled *Cm*TLR5 adopts a single-domain LRR structure that is comprised of 16 complete LRR motifs (LRR1 to LRR16) (Fig 2B). The concave surface is predicted to be smooth where a curved β-sheet structure formed of 16 parallel β strands of LRR modules. As compared to the concave surface, the less regular convex surface is comprised of helices and extended loops. To decipher similarities and differences, we superposed our modeled *Cm*TLR5 with experimental *Dr*TLR5 on corresponding Cα-atom pairs using PyMOL which displayed an RMSD of 0.44 Å (Fig 2C). Structural superposition displayed that most of the LRR motifs perfectly superpose well with each other including the regulatory LRR loop 7 and 9, however, minute differences are observed at the C-terminal end which is mostly helical in *Dr*TLR5. To evaluate the overall stereo chemical quality of our proposed protein model, various protein model evaluation servers were employed. PROCHECK integrated into the SAVeS server evaluated the distribution of (Phi and Psi) angles *via*, Ramachandran plot. The *Cm*TLR5 structure had 99.5% residues in allowed region whereas only 2 residues (*i.e.* Ser253 and Thr189 contributing to 0.5%) fell in disallowed region of the plot indicated that our model fits well with that of experimental structure (Fig 2D). Verify3D analysis of the models revealed that 87.65% of residues had an average 3D-1D score ≥0.2 indicated our modelled structure is quite good. Similarly, the overall quality factor value estimated by ERRAT was 91.85%, which shows that our model is well acceptable for further exercises. Energy profile analysis using ProSA displayed a Z-score of -4.89 (Fig 2E), where the range is typically found for the native

**Table 1. Protein model validation report of *Cm*TLR5.**

| Model validation servers | Parameters employed for protein model evaluation | Model evaluation scores |
|---|---|---|
| **PROCHECK** | Most favored regions (%) | 88.6 |
| | Additionally allowed regions (%) | 10.8 |
| | Generously allowed regions (%) | 0.0 |
| | Disallowed regions (%) | 0.5 |
| | Overall G-factor | 0.08 |
| **ERRAT** | Overall quality | 91.85 |
| **VERIFY 3D** | Averaged 3D-1D score >0.2 | 87.65 |
| **PROQ** | LG score | 4.99 |
| | Max sub score | 0.39 |
| **MOLPROBITY** | Bad backbone bonds (%) | 0.06 |
| | Bad backbone angles (%) | 1.06 |
| | Cβ deviations >0.25 A˚ (%) | 3.67 |
| | Ramachandran outliers | 0.25 |
| **METAMQAPII** | GDT_TS | 47.16 |
| | RMSD | 3.37 |

proteins of similar size in experimental structures. ProQ revealed that *Cm*TLR5 structure was 'very good' based on LG scores (4.99) and MaxSub (0.39). Model validation scores from other validation servers are listed in Table 1. Overall, all of these validation tools strongly suggested that our proposed *Cm*TLR5 model may be accepted as reliable with high confidence. To cross-validate our model, we also computed the secondary structure using PSIPRED server (S2 Fig in S1 File) which perfectly fits with our proposed model.

## MD simulation of *Cm*TLR5

Classical all-atoms molecular dynamics simulations are considered as one of the powerful methods to investigate the structural dynamics and folding under a physiological condition [63, 64]. Moreover, this method is routinely used to investigate the structure, dynamics, and conformational change in macromolecular systems along with ligand binding, etc. [65, 66]. As most proteins associate with other proteins to function, hence, understanding these complexes are central to almost all physiological processes. Using all-atoms long-timescale MD simulations, we explored the recognition of flagellin by *Cm*TLR5 as compared to *Dr*TLR5. The structural dynamics of *Cm*TLR5 alone in an aqueous solution for 200 ns was also extrapolated. The following section demonstrates dynamics of *Cm*TLR5 alone.

The difference between backbone atoms in modeled *Cm*TLR5 conformation was measured through RMSD to the time scale (Fig 3A). From the trend in RMSD, it can be observed that the stability of structures was preserved even after 150 ns of MD simulation. For, *Cm*TLR5 ECD alone, the RMSD value was found to range between 0.23 to 0.27 nm (Fig 3A). The globularity or compactness of the system was measured through $R_g$ which displayed a compact and stable trend with an average value of ~2.67 nm.

Similarly, the Cα-RMSF for *Cm*TLR5 determined the flexibility of each amino acid considering the last 100 ns simulation trajectory. The average fluctuation of modeled *Cm*TLR5 structure persisted below 0.5 nm. One of the noteworthy observations during MD simulations was that residues forming the structural conserved LRR motif are highly flexible with high degree of fluctuation including loop 7 and 9 important for flagellin recognition [14, 18]. The C-terminal (CT) end region was quite flexible than the N-terminal (NT) end region

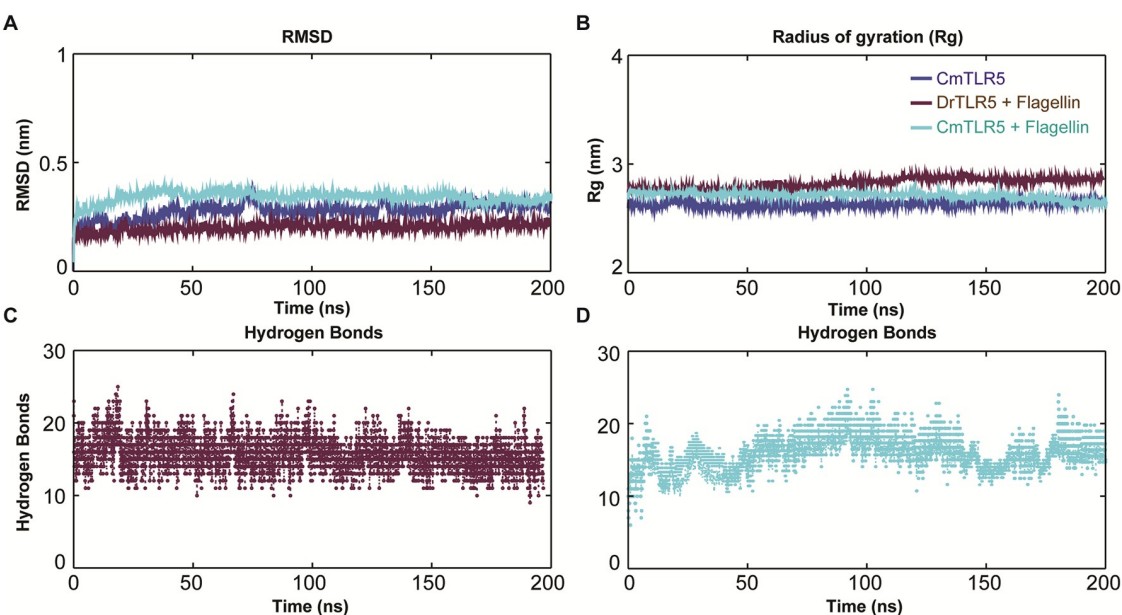

**Fig 3. Intrinsic dynamic stability of the modeled ECD of *Cm*TLR5, *Cm*TLR5-flagellin and *Dr*TLR5-flagellin complex during all-atoms MD simulation of 200 ns.** (**A**) The backbone RMSD of three MD systems over the time scale of 200 ns (Blue: *Cm*TLR5, Maroon: *Dr*TLR5-flagellin and Cyan: *Cm*TLR5-flagellin complex). (**B**) Radius of gyration ($R_g$) plot showing the compactness of trajectory of the three systems over the time scale of 200 ns. (**C**) Variation of intermolecular Hydrogen-bonds forming strong intermolecular contacts of flagellin with *Dr*TLR5 and *Cm*TLR5 during 200 ns MD simulation (**D**).

(Fig 4A). Similarly to corroborate the findings from RMSF analysis, we also computed the b-factors or thermal fluctuations of the modeled *Cm*TLR5 from MD simulation. Our observations from the RMSF analysis perfectly correlates with the b-factor analysis where the central loops displayed a high degree of deformity along the LRR-NT and LRR-CT regions. To understand motion changes of *Cm*TLR5 in more detail, we performed PCA on the MD trajectory using the last 100 ns. To quantitatively comprehend the movement directions captured by the top-ranked two PCs, porcupine plots were generated using the extreme projections on PC1 and PC2 (Fig 4C and 4D). The length of the arrow exemplifies the movement strength while the direction of the arrow embodies the direction of motion. The porcupine plots suggested that LRRCT and LRRNT regions displayed high degree inward motion while the central loop region displayed outward motion in PC1. While in PC2, we could only observe in-ward motion in the LRRCT. The result from PCA perfectly agrees with RMSF and B-factor analysis that the motion modes terminal end and the central loops are the major flexible regions in *Cm*TLR5 ECD.

To investigate the different conformational states of *Cm*TLR5, we performed clustering analysis using gromos method (with a cut-off of 0.2 nm), which uses RMSD based clustering approach. All total, our 10,000 snapshots were clustered into 6 clusters, among which the top two clusters harbors 93.4% of the total snapshots. Therefore, we took the top-ranked representative structures from clustering for structural superposition. The superimposition of the MD simulated top two cluster representatives had an overall RMSD of 1.79 Å (Fig 5A), while with the initial model structure it was 0.79 and 0.82 Å respectively.

Time-dependent variations in the secondary structure elements were subtracted *Cm*TLR5 protein using the whole 200 ns MD trajectory using the VMD timeline utility toolkit (Fig 5B). Overall turns, coil, and loops displayed high degree of flexibility, while, α helices and β strands displayed least-mobile. The evolution of secondary structure element analysis displayed that

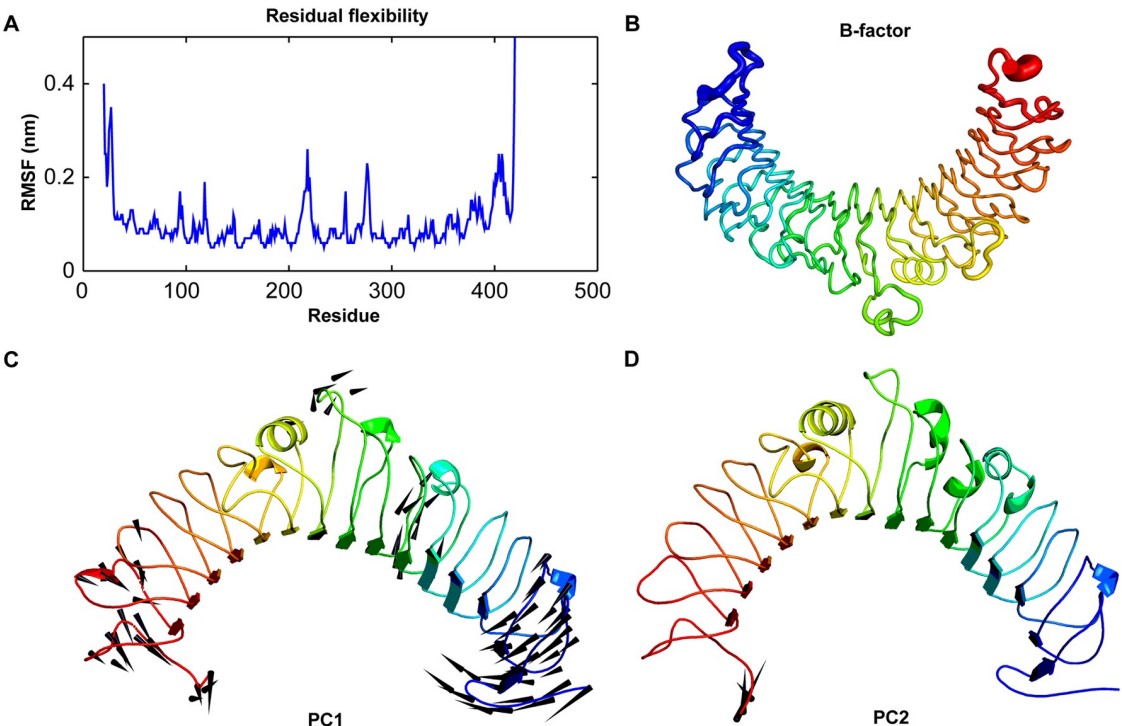

**Fig 4. Residual flexibility and principal component analysis of *Cm*TLR5 alone during the last 100 ns of MD.** (**A**) The residual flexibility is measured by RMSF of Cα-atoms of the *Cm*TLR5. (**B**) B-factor analysis of the MD simulated *Cm*TLR5. (**C**) Porcupine plot displaying the movement of *Cm*TLR5 by analyzing the PC1 from PCA. (**D**) Porcupine plot displaying the movement of *Cm*TLR5 by analyzing the PC2 from PCA.

the loop regions and the helical regions displayed great degree variability during MD; however, the β-strand dominated LRR regions exhibited the least changes.

## Structural dynamics of TLR5-flagellin complexes

To examine changes in the conformation and stability of flagellin bound TLR5 complexes (*i.e.*, *Cm*TLR5-flagellin and *Dr*TLR5-flagellin systems), we performed long term all-atoms MD simulations, as it offers intuitive and vital information on the residues forming inter-molecular contacts in the snapshots of TLR5-flagellin complexes. The conformational dynamics features were analyzed using MD trajectories. RMSD for *Cm*TLR5-flagellin and *Dr*TLR5-flagellin systems were found to be below 3.5 Å during the entire simulation run, where the later had lower RMSD as compared to the former (Fig 3A). The average backbone RMSD for *Cm*TLR5-flagellin complex was found to be ~3.37 Å, while, ~2.23 Å for the *Dr*TLR5-flagellin complex, advocated that at the time of 100 ns, both the protein-protein complex systems attained convergence. The $R_g$ indicates compactness and overall dimension of protein or macromolecular complexes. It enlightens about packing of secondary structures into the 3D structure of a protein [67]. Both the systems displayed a stable and compact $R_g$ profile (Fig 3B). We noticed that the *Dr*TLR5-flagellin complex exhibited an average $R_g$ of ~2.78 nm, while in the *Cm*TLR5-flagellin complex slightly lower $R_g$ of ~2.71 nm was observed indicates more compactness of the system during the 200 ns simulation time. Similar $R_g$ for both the structures suggested tight packing of the TLR-flagellin complexes, making the structure relatively stable (Fig 3B). The solvent-accessible surface area (SASA) for both the systems computed using *gmx sasa* utility toolkit has been summarized in S3 Fig in S1 File.

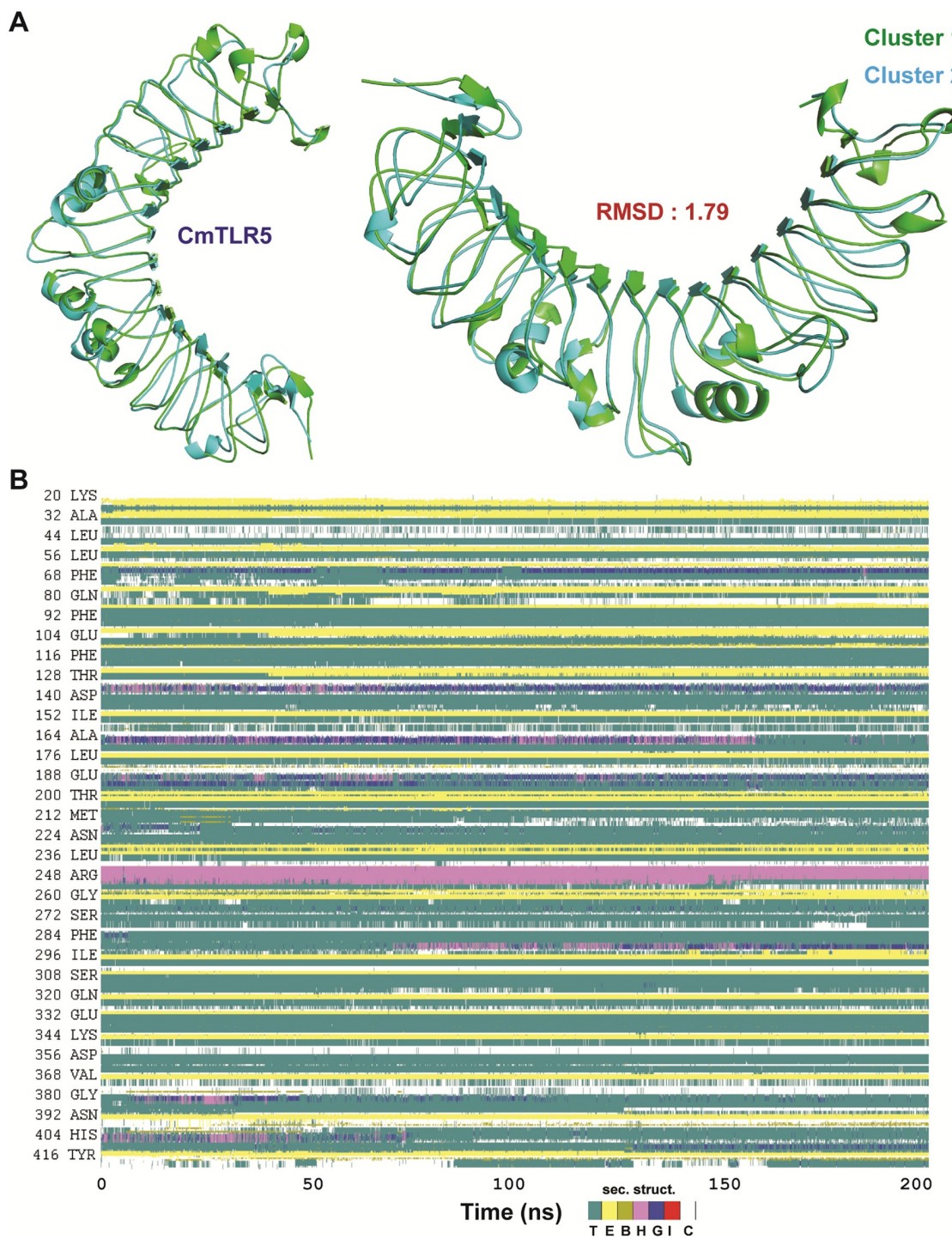

**Fig 5. Structural imposed views of the top two cluster representatives (cluster 1: Green and cluster 2: Blue) obtained from clustering analysis using the snapshots from last 100 ns trajectory (A) and the evolution of secondary structure elements of *Cm*TLR5 over time scale of 200 ns (B).**

The inter-molecular hydrogen bonds (H-bonds) formed between the TLR5 and flagellin were computed by *gmx hbond* utility tool during the entire time of MD simulations. The average H-bonds per time frame for the *Cm*TLR5-flagellin system was computed to be ~17.83 (Fig 3D) while for the *Dr*TLR5-flagellin complex it was computed to be ~15.95 (Fig 3C). Looking at the trend in inter-molecular H-bonds, the *Cm*TLR5-flagellin complex had a slightly variable trend while a comparatively more stable trend was observed in the experimental *Dr*TLR5-flagellin complex.

## Essential dynamics

The collective motion of both of the TLR5-flagellin complexes was explored using the last 100 ns MD trajectories through PCA or essential dynamics. Initially, diagonalization of the covariance matrix using the main-chain atom of the complex was done to capture the strenuous motion of the atoms through eigenvalues and PCs [68]. The eigenvalues signify the atomic contribution of motion while PCs explicate the overall direction of motion of the atoms. To better comprehend the structure and conformational changes, MD trajectories of both the systems were explored in detail [55, 69, 70]. The corresponding eigenvalues showed the dynamic behavior and degree of fluctuation of both the systems (Fig 6A). Eigenvalues of the first few EVs of the complexes take up higher values; however successive eigenvalues were in the decreasing order. The trace value for *Cm*TLR5-flagellin complexes was found to be 25.95 nm$^2$, while for the *Dr*TLR5-flagellin it was 16.88 nm$^2$. Looking at the scatter plot of PC1 vs. PC2 in phase, we observed that the experimental TLR5-flagellin complex occupied the least conformational space as compared to the *Cm*TLR5-flagellin complex. Due to high degree of flexibility, *Cm*TLR5-flagellin complex occupied wider conformational space than the experimental complex (Fig 6B). Further, to understand the global motions in both the systems, we generated porcupine plots using the top two PC1 and PC2. In both the systems, NTR and CTR of TLR5 displayed outward movement while negligible movements were observed in the interacting component flagellin which indicates that later fits well in the cavity formed by the D1 domain for plausible molecular recognition.

## Binding free energy

Qualitative descriptions of the flagellin binding information can be obtained from MD simulation. Further quantitative assessment of the recognition capability of flagellin to the immune receptor is highly essential. Therefore, we attempted to compute the free energy of the binding of both the TLR5-flagellin complexes through the MM/PBSA method. Table 2 summarizes the various energetic terms contributing to binding free energy. Equating the discrete components subsidizing to free energy of binding (Table 2), it can be established that the van der Waals and electrostatic energetic terms govern the change in the strength of binding. In the case of *Cm*TLR5-flagellin complex, we observed a higher contribution in van der Waals and electrostatic terms which may due to the differential binding mediated by charged residues at the binding interface. MM/PBSA approach estimates the binding affinity accurately to some extent; however, due to the non-incorporation of entropic contribution, often it overestimates the absolute free energy of binding [71]. Nevertheless, binding free energy calculations have provided some useful information based on the MD results of TLR5-flagellin complexes. In addition, per-residue free energy decomposition analysis was conducted to obtain more quantitative information on the important residues contributing towards total binding free energy (Fig 7). From Fig 7 it can be observed that the contribution of flagellin residues towards binding free energy is more or less followed the same trend, while the differential contribution is observed in case of TLR5 from zebrafish and mrigala. The major contribution came from the

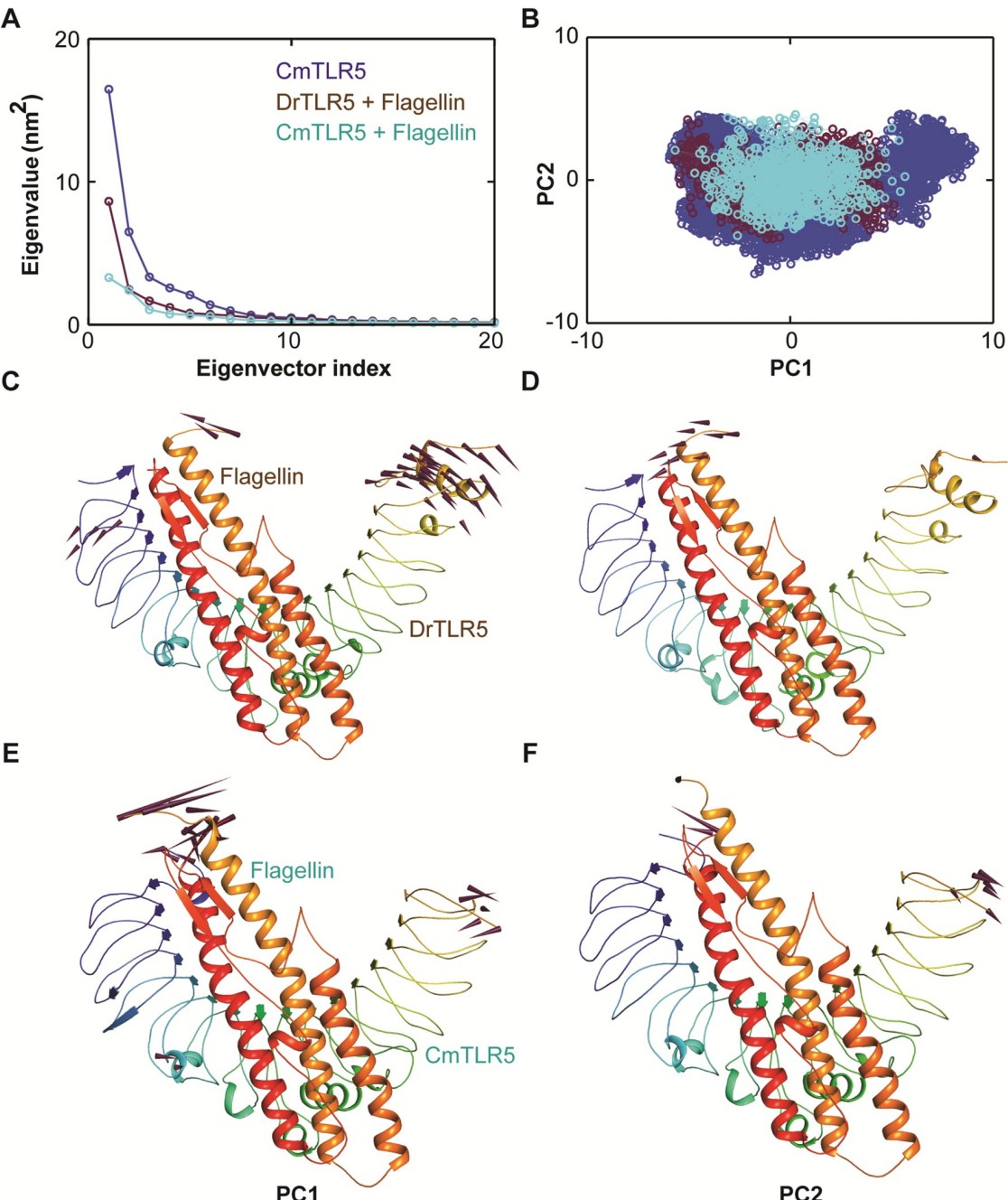

**Fig 6. PCA of *Cm*TLR5-flagellin and *Dr*TLR5-flagellin complexes with using the last 100 ns MD trajectories.** (**A**) Eigenvalues of the few first 20 PCs in TLR5 (alone), *Cm*TLR5-flagellin and *Dr*TLR5-flagellin complexes (**B**) Projection of the motion of the structures of the main-chain atoms in phase space along the first two principal components (PC1 vs. PC2) constructed from the 100 ns MD trajectory of all the systems. (**C**) The Porcupine plot of the first two PCs displaying the movement of the *Dr*TLR5-flagellin complex. (**D**) The Porcupine plot of the first two PCs displaying the movement of the *Cm*TLR5-flagellin complex. The arrows show the tendency of movement.

**Table 2. MM/PBSA binding free energy of the TLR5-Flagellin complex systems obtained using 500 snapshots from the last 100 ns MD trajectories.**

| Energetic component | *Dr*TLR5-Flagellin complex | *Cm*TLR5-Flagellin complex |
|---|---|---|
| van der Waal energy (kJ/mol) | -384.65 ±10.89 | -491.59 ±10.86 |
| Electrostatic energy (kJ/mol) | -669.93±23.26 | -1063.75 ±26.41 |
| Polar solvation energy (kJ/mol) | 884.65±24.20 | 1344.36 ±29.37 |
| SASA energy (kJ/mol) | -51.78±1.42 | -70.79 ±1.60 |
| Binding energy (kJ/mol) | -221.82±10.94 | -282.99 ±10.80 |

central loops which also hold the important residues responsible for molecular recognition [18]. Based on our MD simulation, we observed slightly lower binding affinity for *Dr*TLR5 which may be a combined effect of all amino acid residues.

## Interaction of flagellin with TLR5

To get deep insights into the interactions mediated by flagellin, the top-ranked cluster representative from both the complex systems (S4 Fig in S1 File) were used for interaction analysis in PyMOL and LigPlot[+] [72]. In both the complexes, few common residues *viz*. Glu114, Gln96, and Arg89 of the D1 domain from flagellin formed strong hydrogen bonding with residues of TLR5. In the case of the *Cm*TLR5 complex, a total of 17 H-bonds with an average distance of ~2.83 Å, while in the case of the *Dr*TLR5 11 H-bonds (average distance of ~2.84 Å). Also, major differences exist in the number of hydrophobic contacts where *Cm*TLR5-flagellin complex had more number of electrostatic and pi-alkyl contacts (as shown in Fig 8 and S5 Fig in S1 File) as compared to *Dr*TLR5 complex. This observation perfectly fits with the binding free energy of the complexes. We also measured the inter-atomic distance between the

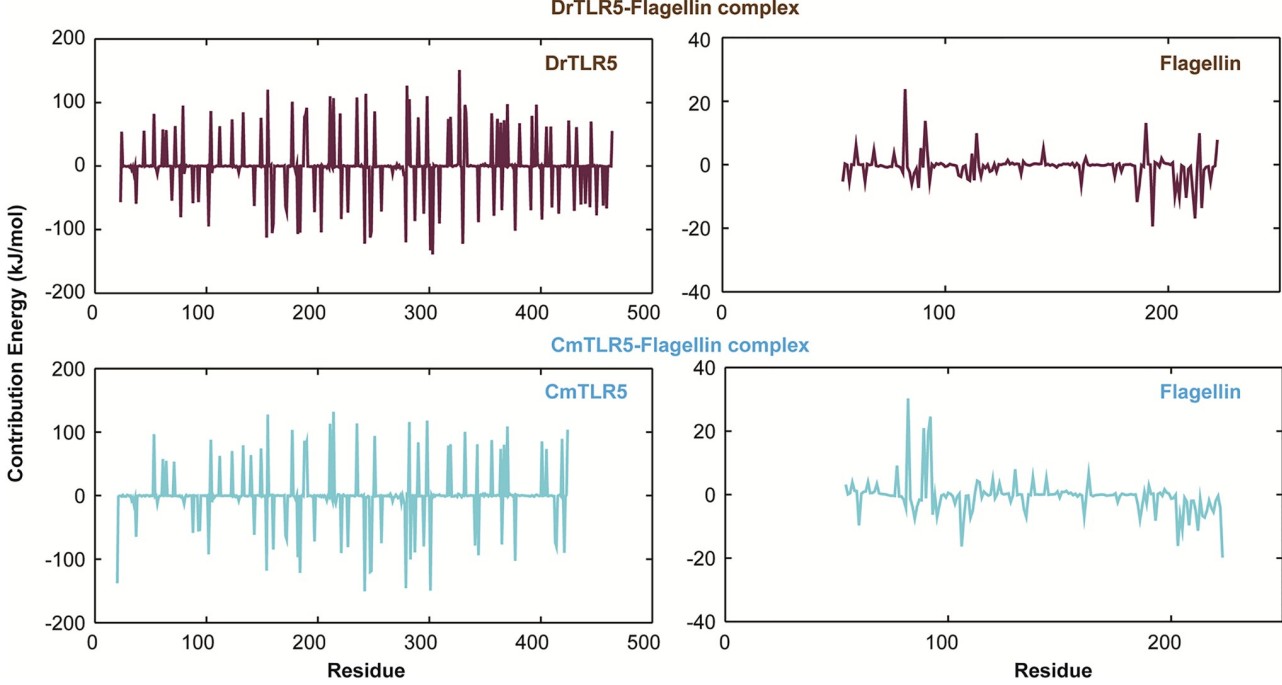

**Fig 7. Per residue decomposition analysis displaying the contribution of each residue towards binding free energy in *Dr*TLR5-Flagellin and *Cm*TLR5-Flagellin complex systems.**

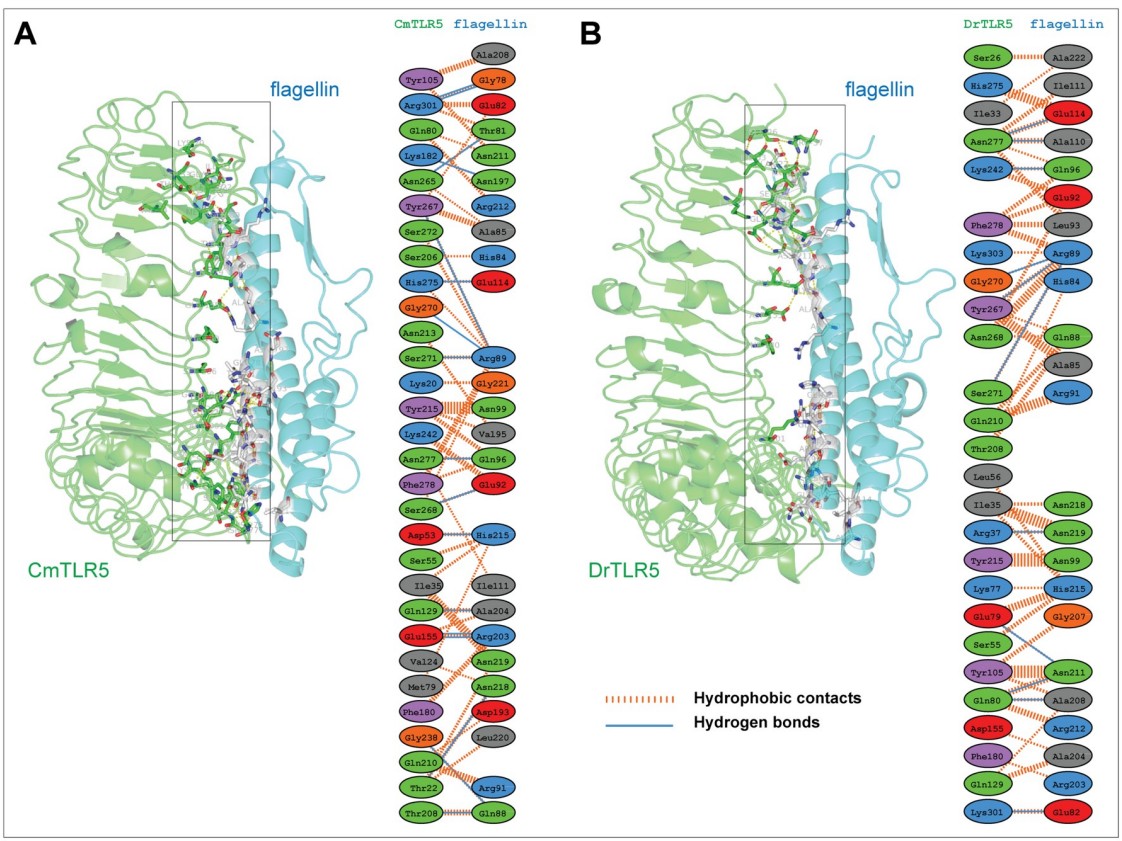

**Fig 8. Protein-protein interaction involving TLR5 and Flagellin obtained from the top-ranked cluster from MD simulations.**
(**A**) Inter-molecular contact analysis of *Cm*TLR5 with flagellin (**B**) Interactions analysis of *Dr*TLR5 with flagellin. The hydrogen bonds are shown in blue line while other hydrophobic non-bonded contacts are displayed in red dashed lines.

important interacting pairs in both the complexes from the MD trajectories (Figs 9 and 10). The average distance between the interacting pairs of atoms from flagellin and TLR5 are summarized in S1 Table in S1 File. The distance profile indicates the continuous participation of these importance amino acid residues in molecular recognition process of flagellin by TLR5. To understand the differences in the contacts, we also computed the electrostatic surface potential map of both the complexes (as summarized in S6 Fig in S1 File). Minute inspection of the binding interface of both complexes revealed that there is a disparity in the distribution of the charged residues which we assume responsibility for this differential binding. The property contact map showed the physiochemical nature of interacting residues of loop 9 regions from TLR5, along with positively charged patch surrounding the Arg90/Arg89 in flagellin has a major role in recognition with TLR5. These results from our study agree with the crystal structure which indicates that Arg90 along with other close by residues of flagellin forms a hot spot at the binding interface between TLR5 and flagellin [18]. Furthermore, alignment of diverse flagellin sequences displayed that residues positioned at 83, 90, and 93 are either charged or polar in nature, which might be responsible for the differential binding of diverse flagellin to TLRs [73]. Overall our study showed that the D1 fragment in the *Cm*TLR5 complex possesses an unusual binding position across the interfaces of recognition. Furthermore, we also affirm that flagellin harbors a number of residues at the primary binding site crucial for species-specific variation in TLR5 recognition [14, 23]. In conclusion, MD simulation of both the TLR5 complexes revealed a stronger flagellin-receptor interaction in the case of mrigala

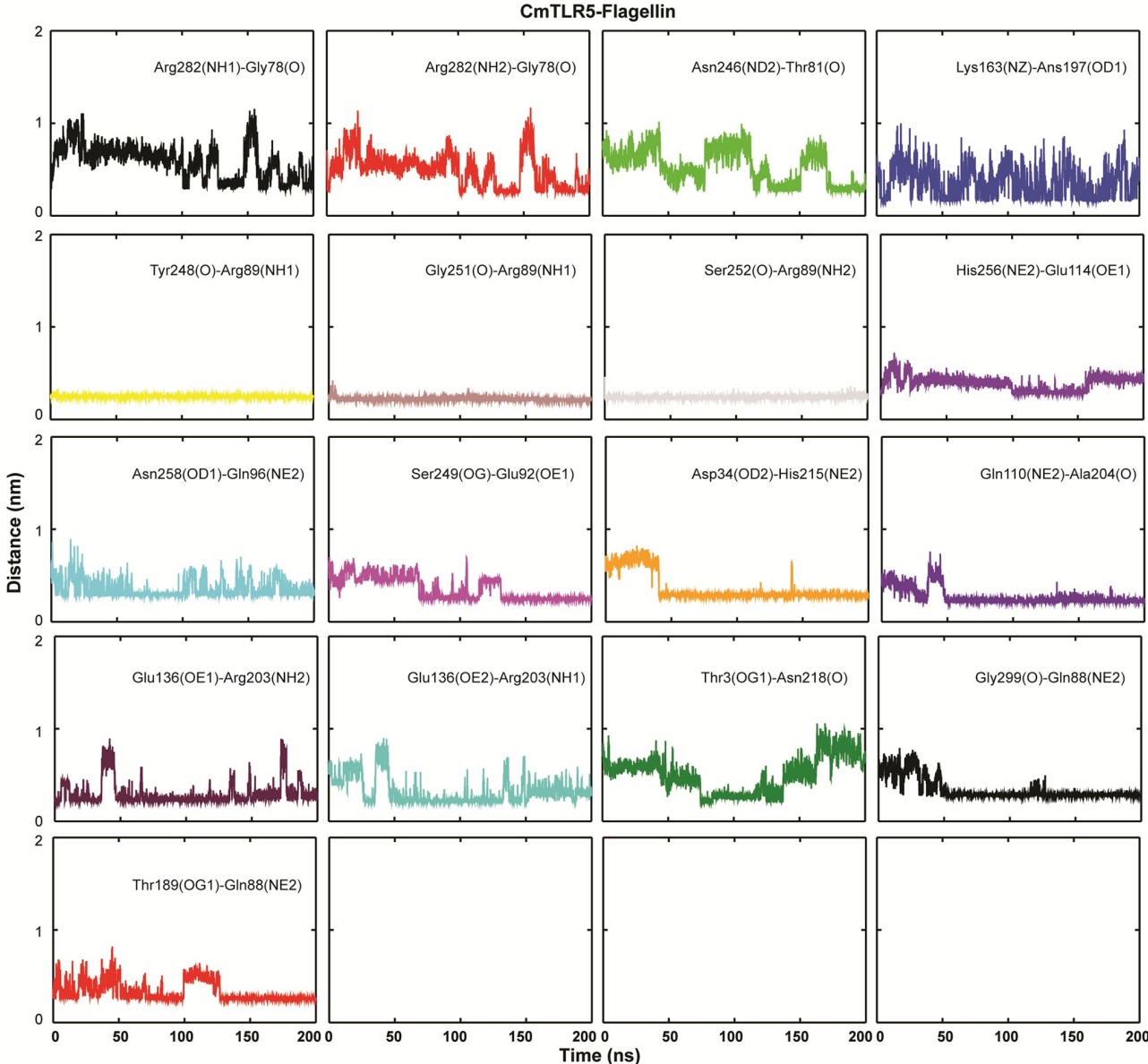

**Fig 9. Inter-atomic distance profile of the important hydrogen bond forming amino acid pairs in *Cm*TLR5-flagellin complex using the MD trajectory.**

and somewhat altered mode of recognition for the region of flagellin recognized by the zebrafish immune receptor TLR5.

## Conclusion

TLR5 plays a significant role in the innate immune system, host defense, and bacterial PAMP recognition involved in signal transduction. In teleosts, TLR5 exists in multiple copies and plays a pivotal role in the development of fishes. However, due to inadequate biochemical and structural information on the extracellular domain of TLR5 in fishes, little is known about their interactions with bacterial components including flagellins. Here, the present investigation seeks to understand the structural dynamics of TLR5 from major carp mrigala and its

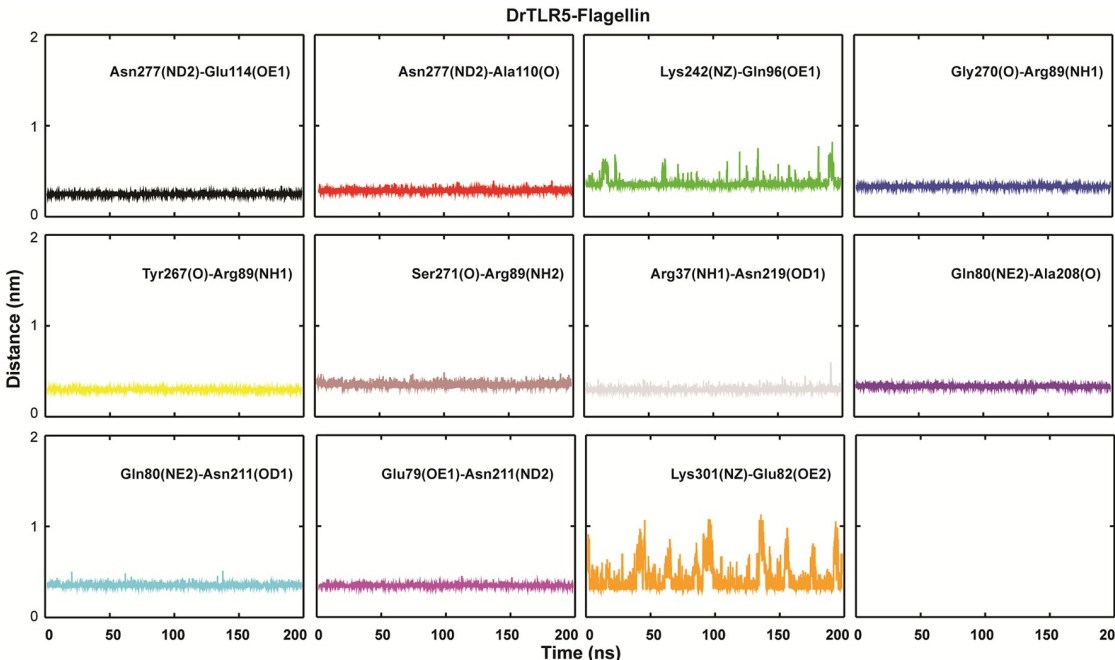

**Fig 10. Inter-atomic distance profile of the important hydrogen bond forming amino acid pairs in *Dr*TLR5-flagellin complex using the 200 ns MD trajectory.**

interaction with Salmonella flagellin. To summarize, our *in-silico* findings based on conformational ensembles and mode of flagellin recognition, D1 domain of flagellin recognizes LRR motifs of *Cm*TLR5 in aqueous solution. TLR5-flagellin receptor complexes in zebrafish and mrigala possess conformational stability when the extracellular LRR domain recognizes the D1 domain of flagellin. The mode of recognition takes place through D1 of flagellins; however, we observe binding interfaces are unusual due to differential distribution of charged residues in the binding interface. In both TLR5-flagellin complexes, van der Waals and electrostatic interactions are the major driving force behind the molecular recognition of flagellins. Our results perfectly corroborate with earlier studies which suggest that most mobile and disordered residues of flagellin are responsible for the differential binding of bacterial components like flagellin to a cognate immune receptor. Further studies involving other PAMPs from eukaryotic system will shed more insights into the molecular recognition of diverse flagellins in near future. Moreover, comparative structural dynamics studies supported by site directed mutational analyses will be useful for the atomic understanding of flagellin mediated interaction from diverse bacterial organisms with host immune receptor, and thus have potential application in pharmaceutical developments.

## Supporting information

**S1 File. The supplementary data to this article can be found in the supporting information.** (DOCX)

## Acknowledgments

The authors thank Mr. Asim Kumar Jana, Senior Technical Assistant, ICAR-Central Inland Fisheries Research Institute, Barrackpore, Kolkata, India.

## Author Contributions

**Conceptualization:** Pranaya Kumar Parida, Bijay Kumar Behera.

**Data curation:** Ajaya Kumar Rout, Budheswar Dehury.

**Formal analysis:** Varsha Acharya, Diptimayee Maharana, Budheswar Dehury.

**Investigation:** Bhaskar Behera, Pranaya Kumar Parida, Bijay Kumar Behera.

**Software:** Ajaya Kumar Rout, Budheswar Dehury.

**Validation:** Ajaya Kumar Rout, Sheela Rani Udgata, Rajkumar Jena.

**Writing – original draft:** Ajaya Kumar Rout, Varsha Acharya, Diptimayee Maharana, Budheswar Dehury, Sheela Rani Udgata, Rajkumar Jena, Bhaskar Behera, Pranaya Kumar Parida, Bijay Kumar Behera.

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
