## [Decision Letter · Decision Letter 0]

10 Dec 2020

PONE-D-20-36156

Insights into structure and dynamics of extracellular domain of Toll-like receptor 5 in Cirrhinus mrigala (mrigala): A molecular dynamics simulation approach

PLOS ONE

Dear Dr. Behera,

Thank you for submitting your manuscript to PLOS ONE. After careful consideration, we feel that it has merit but does not fully meet PLOS ONE’s publication criteria as it currently stands. Therefore, we invite you to submit a revised version of the manuscript that addresses the points raised during the review process.

We look forward to receiving your revised manuscript.

Kind regards,

Nagendra Kumar Kaushik, PhD

Academic Editor

PLOS ONE

Journal Requirements:

Additional Editor Comments:

Author must revise manuscript according to both reviewers comments.

Reviewers' comments:

Reviewer's Responses to Questions

**Comments to the Author**

1. Is the manuscript technically sound, and do the data support the conclusions?

Reviewer #1: Yes

Reviewer #2: Yes

2. Has the statistical analysis been performed appropriately and rigorously? 

Reviewer #1: Yes

Reviewer #2: Yes

3. Have the authors made all data underlying the findings in their manuscript fully available?

Reviewer #1: Yes

Reviewer #2: Yes

4. Is the manuscript presented in an intelligible fashion and written in standard English?

Reviewer #1: Yes

Reviewer #2: Yes

5. Review Comments to the Author

Reviewer #1: The submitted manuscript entitled “Insights into structure and dynamics of the extracellular domain of Toll-like receptor 5 in Cirrhinus mrigala (mrigala): A molecular dynamics simulation approach” illustrates the binding of flagellin to TLR5-ECD from major carp mrigala. The authors have used standard molecular modeling and all-atoms MD simulation to achieve their defined objectives. The methodology sounds good. There are some comments before consideration of publication.

The authors should include the latest updates on TLR5 from fishes as well as other homologs. It would be suggested to authors to calculate and report the distance (minimum distance) among the important interacting pairs of residues in both CmTLR5-flagellin and DrTLR5-flagellin trajectories. The discussion section needs improvement.

Reviewer #2: The research article entitled “Insights into structure and dynamics of extracellular domain of Toll-like receptor 5 in Cirrhinus mrigala (mrigala): A molecular dynamics simulation approach” describes the differential binding of flagellin to the ECD of TLR5 from mrigal, a major carp through molecular modeling and MD simulation. The article is well written and the objectives are clearly defined. The methodologies adopted in this study are well defined. MS may be accepted after minor revision.

1. The authors should include a multiple sequence alignment of ECD from TLR5 including mrigala with its close homologs displaying the conserved and variable regions.

2. The Introduction section should be modified and the latest references need to be included.

3. The authors may represent the distance analysis of both DrTLR5-flagellin and CmTLR5-flagellin.

4. The authors may describe elaborately Figure 1 - the phylogeny tree

5. The authors should also discuss the findings of their study with the existing reports

6. The conclusion has to be more concise.

6. PLOS authors have the option to publish the peer review history of their article (what does this mean?). If published, this will include your full peer review and any attached files.

Reviewer #1: No

Reviewer #2: No

---

## [Author Response · Author response to Decision Letter 0]

22 Dec 2020

Response to Reviewer’s 

We would to like thank the peer reviewers for positive their feedback and valuable suggestion on our MS (PONE-20-36156). We have tried our best to address each comment raised by the reviewers. Please find the point by point response to each comment below.

Reviewer #1

Comment

The authors should include the latest updates on TLR5 from fishes as well as other homologs. 

Response

As per the reviewer’s suggestion, we have performed a multiple sequence alignment of the ECD of TLR5 from mrigala will its close homologs obtained after BLAST against NR database of NCBI (S1Fig). 

Comment

It would be suggested to authors to calculate and report the distance (minimum distance) among the important interacting pairs of residues in both CmTLR5-flagellin and DrTLR5-flagellin trajectories. 

Response

Thank you so much for raising this important question. We have computed the distance between the interacting pairs (the Hydrogen-bond forming residues in TLR5 and Flagellin) in both the complexes using the MD trajectories (as shown in Figure 9 and 10). We have also included a paragraph describing the important interacting pairs in both the complexes (Page no: 27 and paragraph: Interaction of Flagellin with TLR5) and the average distance has been present in Table S1.

Comment

The discussion section needs improvement.

Response

The discussion section of the MS has been revised as per the suggestion.

Reviewer #2

Comment 1

The authors should include a multiple sequence alignment of ECD from TLR5 including mrigala with its close homologs displaying the conserved and variable regions.

Response

As per the reviewer’s suggestion, we have included a multiple sequence alignment of the ECD from TLR5 of mrigala with its close homologs. The alignment displays the most conserved and variable region in the ECD of TLR5 from diverse species (S1 Fig). 

Comment

The Introduction section should be modified and the latest references need to be included.

Response

The Introduction has been revised keeping the suggestion of the peer reviewer with latest research articles.

Comment

The authors may represent the distance analysis of both DrTLR5-flagellin and CmTLR5-flagellin.

Response

As per the reviewer recommendation we have performed distance analysis of both the complexes (as summarized in Figure 9 & 10 and Table S1).

Comment 4

The authors may describe elaborately Figure 1 - the phylogeny tree

Response

In the revised MS, we have discussed the findings from phylogenetic analysis as shown in Figure 1.

Comment

The authors should also discuss the findings of their study with the existing reports

Response

The discussion part has been revised as suggested by the Reviewer and incorporated in the manuscript.

Comment

The conclusion has to be more concise.

Response

As per the reviewers comment we have revised the conclusion section in the manuscript.

---

## [Editor Report · Decision Letter 1]

29 Dec 2020

Insights into structure and dynamics of extracellular domain of Toll-like receptor 5 in Cirrhinus mrigala (mrigala): A molecular dynamics simulation approach

PONE-D-20-36156R1

Dear Dr. Behera,

We’re pleased to inform you that your manuscript has been judged scientifically suitable for publication and will be formally accepted for publication once it meets all outstanding technical requirements.

Kind regards,

Nagendra Kumar Kaushik, PhD

Academic Editor

PLOS ONE

Additional Editor Comments (optional):

I recommend accepting this manuscript as author already revised manuscript properly.

---

## [Editor Report · Acceptance letter]

6 Jan 2021

PONE-D-20-36156R1 

Insights into structure and dynamics of extracellular domain of Toll-like receptor 5 in *Cirrhinus mrigala* (mrigala): A molecular dynamics simulation approach 

Dear Dr. Behera:

I'm pleased to inform you that your manuscript has been deemed suitable for publication in PLOS ONE. Congratulations! Your manuscript is now with our production department. 

Kind regards, 

on behalf of

Prof. Nagendra Kumar Kaushik 

Academic Editor

PLOS ONE